# International patient preferences for physician attire: results from cross-sectional studies in four countries across three continents

Nathan Houchens [1,2] Sanjay Saint,[1,2] Christopher Petrilli,[3] Latoya Kuhn,[1,2] David Ratz,[1] Lindsey De Lott,[4] Marc Zollinger,[5] Hugo Sax,[6] Kazuhiro Kamata,[7,8] Akira Kuriyama,[9] Yasuharu Tokuda [10] Carlo Fumagalli [11] Gianni Virgili [12,13] Stefano Fumagalli,[11] Vineet Chopra[14]

For numbered affiliations see end of article.

**Correspondence to**
Dr Nathan Houchens;
nathanho@med.umich.edu

## ABSTRACT

**Objective** The patient–physician relationship impacts patients' experiences and health outcomes. Physician attire is a form of nonverbal communication that influences this relationship. Prior studies examining attire preferences suffered from heterogeneous measurement and limited context. We thus performed a multicentre, cross-sectional study using a standardised survey instrument to compare patient preferences for physician dress in international settings.

**Setting** 20 hospitals and healthcare practices in Italy, Japan, Switzerland and the USA.

**Participants** Convenience sample of 9171 adult patients receiving care in academic hospitals, general medicine clinics, specialty clinics and ophthalmology practices.

**Primary and secondary outcome measures** The survey was randomised and included photographs of a male or female physician dressed in assorted forms of attire. The primary outcome measure was attire preference, comprised of composite ratings across five domains: how knowledgeable, trustworthy, caring and approachable the physician appeared, and how comfortable the respondent felt. Secondary outcome measures included variation in preferences by country, physician type and respondent characteristics.

**Results** The highest rated forms of attire differed by country, although each most preferred attire with white coat. Low ratings were conferred on attire extremes (casual and business suit). Preferences were more uniform for certain physician types. For example, among all respondents, scrubs garnered the highest rating for emergency department physicians (44.2%) and surgeons (42.4%). However, attire preferences diverged for primary care and hospital physicians. All types of formal attire were more strongly preferred in the USA than elsewhere. Respondent age influenced preferences in Japan and the USA only.

**Conclusions** Patients across a myriad of geographies, settings and demographics harbour specific preferences for physician attire. Some preferences are nearly universal, whereas others vary substantially. As a one-size-fits-all dress policy is unlikely to reflect patient desires and expectations, a tailored approach should be sought that attempts to match attire to clinical context.

## STRENGTHS AND LIMITATIONS OF THIS STUDY

⇒ With over 9000 participants, this is the largest international study examining opinions on physician dress to date.

⇒ Use of a standardised survey instrument allowed direct comparisons across diverse geographic regions, populations, physician types and clinical contexts.

⇒ Robust and careful survey design, including randomisation and constant photographic features, mitigated bias and confounding.

⇒ Comparative over-representation of the USA and convenience sampling may have contributed to disproportionate representation.

⇒ The survey instrument used predefined Likert scales, which may not accurately reflect nuanced patient opinions, and which do not capture other elements of patient–physician interactions.

## INTRODUCTION

Successful patient–physician relationships are founded on mutual respect, trust, confidence and care. The strength of these connections can directly impact patients' experiences with healthcare, satisfaction and important health outcomes such as adherence to treatment recommendations,[1 2] 30-day readmissions[3] and mortality.[4] Patient–physician interactions are complex and dependent on multiple factors including social definitions and cultural norms. To ensure the highest quality care, it is essential to identify techniques that physicians may use to establish and maintain strong relationships with their unique individual patients while recognising the influence of sociocultural context. From initial introductions, physicians employ verbal and nonverbal communication to form impressions and cultivate partnerships with their patients.[5]

The clothing worn by a physician is one form of nonverbal communication that may influence the patient–physician relationship. Physician attire is an important element in establishing patient confidence and trust,[6] enhancing patient comfort when discussing personal problems[7–9] and shaping patient perceptions of physician professionalism,[6] intelligence[10] and empathy.[11] Most prior scholarship has focused on a single geographic region, country or clinical context (eg, primary care clinic, hospital setting)[12–15] and has not considered the relative impacts of different physician specialties, contexts of care, geography and patient factors such as age, education and gender. In addition, heterogeneity among prior studies, such as different sampling methodology and survey instruments, has made comparisons across different studies challenging.

The objective of this international, multicentre, cross-sectional study was to use a structured survey instrument to examine patient preferences for physician attire in different regions, countries and continents. The survey instrument allowed direct comparisons among a variety of cultures and contexts, thereby mitigating the heterogeneity of prior studies.[16–18] We report comparisons of data from five primary cross-sectional survey research studies conducted in Italy, Japan,[19] Switzerland[20] and the USA.[21 22] Our aim was to identify common themes and differences of patient expectations for physician dress so that we may tailor attire and thus elevate the patient experience and optimise health outcomes.

## METHODS
### Study design and participants
We performed a survey-based study using a convenience sample of patients in 20 hospitals and healthcare practices in Italy, Japan, Switzerland and the USA. These sites were selected based on our research networks and availability of clinicians who would serve as leads in their respective institutions. Sites included academic hospitals (general medicine wards, intensive care units), general medicine ambulatory clinics, specialty ambulatory clinics (dermatology, infectious disease, neurology, orthopaedic surgery) and ophthalmology practices (table 1). Data collection took place between June 2015 and October 2017.

At each participating healthcare location, the research team printed and randomly administered a survey instrument, targeting representative adult patients who were receiving clinical care at one of those sites. Participants were presented with a paper-based instrument of 22 questions that included photographs of either a male or female physician wearing various forms of attire and asked to rate their preferences. Respondents could request assistance with form completion from persons accompanying them.

All participants provided informed verbal consent. No identifying information was collected from participants who completed the study. Institutional permission for recruitment and data collection was obtained from each site.

### Procedures
The 22-item survey instrument was developed following a systematic review of the literature that examined the role of physician attire on the patient experience.[23] The survey instrument was developed and piloted by a multidisciplinary team to gather feedback and refine photographs, questions, rating scale, presentation order and randomisation scheme. Questions were translated into different languages for each country by interpreters at each site: Italian for Italy, Japanese for Japan, German for Switzerland (since the Swiss survey was conducted in Zurich), and English for the USA.

**Table 1** Characteristics of participating study sites

| Country | Dates of data collection | Types of outpatient clinics | Clinical setting(s) | Hospitals, Practices | Geographic regions sampled | Surveys completed |
|---|---|---|---|---|---|---|
| Italy | 10/26/2015-10/21/2016 | Infectious Disease, Ophthalmology, Geriatric Intensive Care Unit | Outpatient and Inpatient | 1 | 1* | 958 |
| Japan | 12/01/2015-10/30/2017 | General Medicine, Medicine Specialties, Orthopaedic Surgery | Outpatient and Inpatient | 4 | 3† | 2020 |
| Switzerland | 06/15/2015-10/31/2016 | Dermatology, Infectious Disease, Neurology | Outpatient | 1 | 1‡ | 834 |
| USA§ | 06/01/2015-10/31/2016 | General Medicine, Medicine Specialties | Outpatient and Inpatient | 10 | 4¶ | 4062 |
| | | Ophthalmology | Outpatient | 4 | 3** | 1297 |

*One site in the Tuscany region.
†Two sites in the Kantō region; one site in the Kansai region; one site in the Chūgoku region.
‡One site in the Canton of Zurich.
§Geographic regions of the USA include Northeast, Midwest, South and West.
¶Three sites in the Midwest, three sites in the South, two sites in the Northeast, two sites in the West.
**Two sites in the Midwest, one site in the Northeast, one site in the West.

Each question referenced particular preferences and opinions of respondents in relation to photographs of medical providers wearing seven unique forms of attire. The forms of dress presented included: casual, casual with white coat, scrubs, scrubs with white coat, formal, formal with white coat and business suit. Photographs were taken with attention paid to achieving constant physician facial expressions as well as consistent visual cues such as lighting, background and pose. Photographs used at all study sites were identical with the following exceptions: In Switzerland, photographs of physicians in medical attire contextually appropriate to the Swiss health system (ie, white scrubs instead of blue scrubs) were used. All other photographic elements including physician models and other forms of attire were unchanged. In Japan, photographs of physicians of Japanese descent with slightly modified attire were used (online supplemental appendix A).

Each survey instrument had four sections. The first section showed a photograph of either a male or female physician wearing one of the seven unique forms of attire. To avoid biases such as anchoring, priming, order response, and gender conformity, 14 different versions of the survey instrument were created. The gender and attire of the first photograph seen by each respondent were randomised; all other sections of the survey were identical (online supplemental appendix B).

## Measurements

Respondents were first asked to rate the standalone, randomised physician photograph using a 1–10 scale across five domains (ie, how knowledgeable, trustworthy, caring and approachable the physician appeared, and how comfortable the physician's appearance made the respondent feel), with a score of 10 representing the highest rating. Respondents were subsequently given seven photographs of the same physician wearing various forms of attire. Questions were asked regarding preference of attire in varied clinical settings (ie, primary care, emergency department, hospital, surgery) and overall preference. To identify the influence of and respondent preferences for physician dress and white coats, a Likert scale ranging from 1 (strongly disagree) to 5 (strongly agree) was employed. Preferences for attire by respondent characteristics such as age, gender, education level, nationality and number of unique physicians seen in the past year were collected. Unanswered questions and those with multiple responses were excluded.

The primary outcome of attire preference was calculated as the mean composite score of the five individual rating domains (ie, knowledgeable, trustworthy, caring, approachable and comfortable), with the highest score representing the most preferred form of attire. We also assessed variation in preferences for physician attire between countries, by physician type and clinical setting, and by respondent characteristics such as age and gender.

## Statistical analysis

Survey data were entered independently and in duplicate by the study teams. Respondents were not required to answer all questions; therefore, the denominator for each question varied. Data were reported as mean and SD, or N and percentage, where appropriate. Differences in the mean composite rating scores between countries were assessed using one-way analysis of variance (ANOVA) with Tukey's method for pairwise comparisons. Differences in mean composite score within country by sociodemographic factors were assessed using Student's t-tests. Differences between countries with respect to categorical responses were compared by using $\chi^2$ tests. Statistical tests were assessed using $p<0.05$ considered significant. All analyses were performed using SAS V9.4 (SAS).

## Patient and public involvement

Patients were not included in the design of the survey instrument, recruitment or conduct of the study. Patients who participated did so anonymously, and therefore, the study team will be unable to disseminate the results to study participants.

## RESULTS

### Characteristics of study sites and participants

A total of 9171 patients completed the survey instrument in outpatient and inpatient healthcare settings within a total of 20 hospitals or practices, 13 distinct geographic regions, 4 countries and 3 continents. Patients were examined in age ranges of 18–64 years and 65 years or older. Patients aged 65 years or older comprised 36.0% of all respondents with substantial age variation across countries. For instance, those 65 years or older represented 48.5% of respondents in Japan, 35.6% in the USA, 27.8% in Italy and 16.7% in Switzerland. Among all respondents, 44.9% were female, 39.6% had a college or graduate degree and 26.6% had seen 6 or more physicians in the previous year. Characteristics of study sites are found in table 1, and sociodemographic characteristics of respondents are described in table 2.

### Ratings of attire types by country

Responses regarding patient preferences for physician attire varied by country. Formal attire with white coat received the highest ratings from respondents in Italy and the USA with mean composite scores of 7.5 (SD 1.8) and 8.1 (SD 1.8), respectively. Conversely, scrubs with white coat received the highest ratings in Switzerland (mean composite score of 7.5 (SD 1.7)) and casual attire with white coat in Japan (mean composite score of 7.1 (SD 1.8)). The forms of attire that received the lowest mean composite ratings were business suit in Italy, Japan, and Switzerland with mean composite scores of 5.6 (SD 2.4), 5.5 (SD 2.1) and 5.2 (SD 2.2), respectively, and casual attire in the USA with a mean composite score of 6.2 (SD 2.5). Within each country, composite scores for attire forms with white coat were higher than those for

**Table 2** Sociodemographic information

| | Italy (n=958) | Japan (n=2020) | Switzerland (n=834) | USA (n=5359) | Total (n=9171) |
|---|---|---|---|---|---|
| Age | n=928 | n=2010 | n=812 | n=5279 | n=9029 |
| 18–25 | 61 (6.6%) | 67 (3.3%) | 50 (6.2%) | 241 (4.6%) | 419 (4.6%) |
| 26–34 | 89 (9.6%) | 162 (8.1%) | 93 (11.5%) | 464 (8.8%) | 808 (9.0%) |
| 35–54 | 310 (33.4%) | 461 (22.9%) | 341 (42.0%) | 1299 (24.6%) | 2411 (26.7%) |
| 55–64 | 210 (22.6%) | 345 (17.2%) | 192 (23.6%) | 1393 (26.4%) | 2140 (23.7%) |
| ≥65 | 258 (27.8%) | 975 (48.5%) | 136 (16.7%) | 1882 (35.6%) | 3251 (36.0%) |
| Gender | n=905 | n=2011 | n=806 | n=5194 | n=8916 |
| Female | 471 (52.0%) | 1040 (51.7%) | 304 (37.7%) | 2184 (42.0%) | 3999 (44.9%) |
| Male | 434 (48.0%) | 971 (48.3%) | 502 (62.3%) | 3010 (58.0%) | 4917 (55.1%) |
| Education | n=919 | n=2010 | n=808 | n=5247 | n=8984 |
| Less than high school | 237 (25.8%) | 243 (12.1%) | 368 (45.5%) | 146 (2.8%) | 994 (11.1%) |
| High school diploma | 416 (45.3%) | 1236 (61.5%) | 82 (10.2%) | 2691 (51.3%) | 4425 (49.3%) |
| College degree | 77 (8.4%) | 487 (24.2%) | 340 (42.1%) | 1490 (28.4%) | 2394 (26.6%) |
| Graduate degree | 189 (20.5%) | 44 (2.2%) | 18 (2.2%) | 920 (17.5%) | 1171 (13.0%) |
| No of unique physicians seen in the past year | n=928 | n=2009 | n=810 | n=5265 | n=9012 |
| 0 | 76 (8.2%) | 38 (1.9%) | 13 (1.6%) | 51 (1.0%) | 178 (2.0%) |
| 1 | 126 (13.6%) | 140 (7.0%) | 83 (10.2%) | 377 (7.2%) | 726 (8.1%) |
| 2 | 199 (21.4%) | 373 (18.5%) | 165 (20.4%) | 769 (14.6%) | 1506 (16.7%) |
| 3 | 188 (20.3%) | 512 (25.5%) | 203 (25.1%) | 940 (17.9%) | 1843 (20.4%) |
| 4 | 112 (12.1%) | 359 (17.9%) | 126 (15.6%) | 824 (15.6%) | 1421 (15.8%) |
| 5 | 84 (9.0%) | 225 (11.2%) | 57 (7.0%) | 571 (10.8%) | 937 (10.4%) |
| ≥6 | 143 (15.4%) | 362 (18.0%) | 163 (20.1%) | 1733 (32.9%) | 2401 (26.6%) |

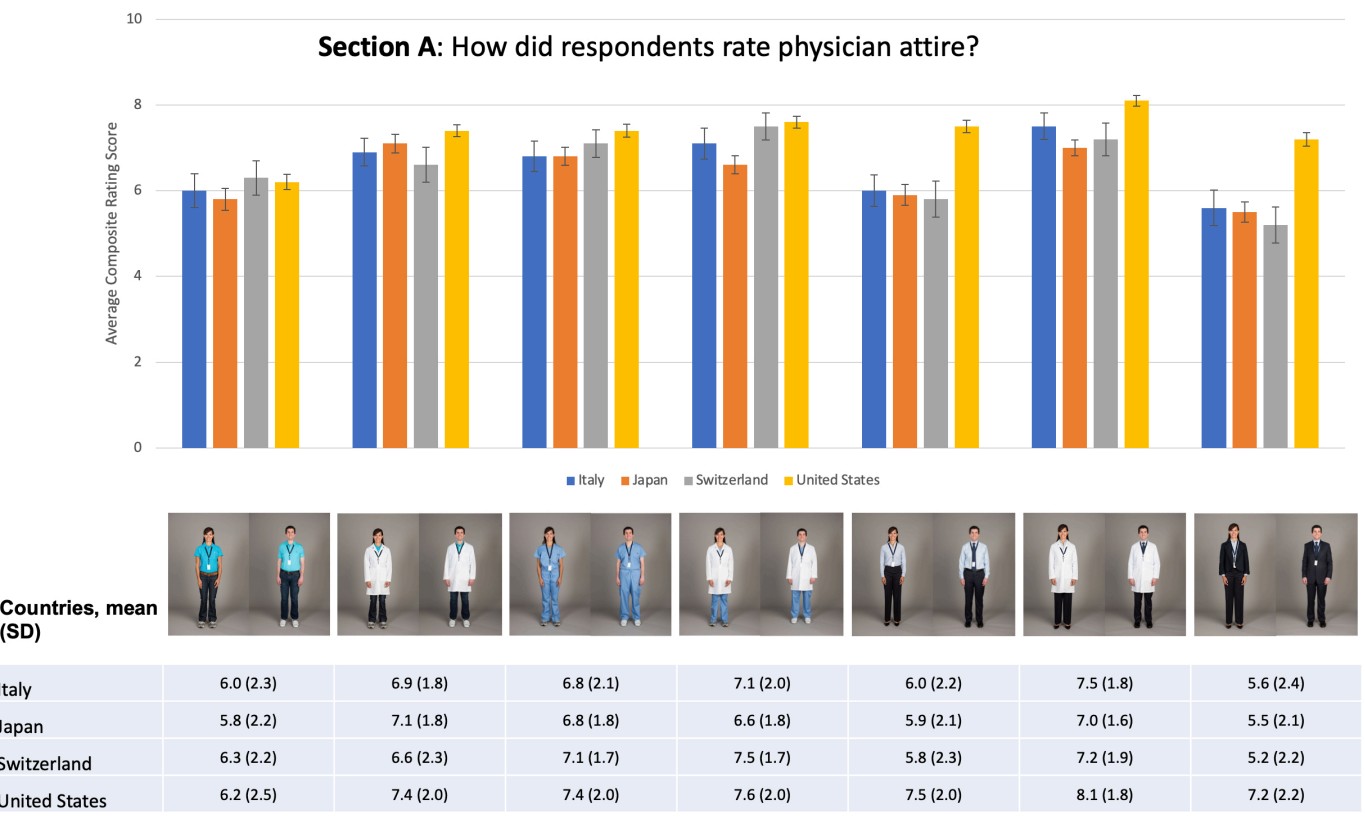

**Figure 1** Mean composite ratings of physician attire.

the corresponding forms without white coat, with only one exception (composite scores for scrubs and scrubs with white coat in Japan were 6.8 and 6.6, respectively). Ratings of different forms of attire by country are found in figure 1 and ratings of physician attire by domain are found in online supplemental appendix C.

## Comparisons of patient preferences between countries
### Preferences for physician attire by type of attire

Similarities between countries when comparing preferences for different types of physician attire were observed. For instance, there was complete concordance for all types of attire between the European countries of Italy and Switzerland. There was near complete concordance when comparing Italy and Japan, with the only statistically significant difference of Italy more strongly preferring formal attire with white coat compared with Japan (mean composite rating difference 0.54, simultaneous 95% confidence limits 0.06 to 1.01). Similarly, there was near complete concordance when comparing Switzerland and Japan, with the only significant difference of Switzerland more strongly preferring scrubs with white coat compared with Japan (mean composite rating difference 0.90, simultaneous 95% confidence limits 0.36 to 1.44). Among all types of attire, the form with the most concordance across countries was casual attire, with no between-country differences identified.

Just as ratings for physician attire varied by country, preferences for specific forms of attire also differed. For instance, the USA significantly more strongly preferred both forms of scrubs-based attire when compared with Italy and Japan, but not when compared with Switzerland. In addition, the USA significantly more strongly preferred all forms of formal attire (ie, formal attire with or without white coat and business suit) when compared with the other countries. These results are summarised in online supplemental appendix D.

### Preferences for physician attire by type of physician

Photographs of either a male or female physician in seven different forms of attire (online supplemental appendix B) were shown, and respondents were asked to select which attire they preferred for different physician types. With respect to primary care physicians, respondents had varying preferences for attire. The highest rated forms in each country were formal attire with white coat in Italy (31.6%) and the USA (46.8%), casual attire with white coat in Japan (34.1%) and casual attire in Switzerland (24.4%). Heterogeneity in patient preferences was particularly noted in Switzerland with nearly equal preference given to casual attire, casual attire with white coat and formal attire with white coat. The highest rated form of attire across all respondents was formal attire with white coat (40.1%).

With respect to hospital-based physicians, preferences again diverged. The highest rated forms in each country were scrubs with white coat in Italy (43.8%) and Switzerland (35.0%), casual attire with white coat in Japan

(34.0%) and formal attire with white coat in the USA (37.6%). The highest rated form of attire across all respondents was formal attire with white coat (32.8%).

With respect to both emergency department physicians and surgeons, preferences were more uniform. Among all respondents across all countries, the most preferred form of attire was scrubs (44.2% for emergency department physicians, 42.4% for surgeons) followed by scrubs with white coat (30.4% for emergency department physicians, 25.4% for surgeons).

With respect to the most preferred form of attire overall, differences between countries were noted. The top forms of attire in each country were scrubs with white coat in Italy (41.7%) and Switzerland (31.5%) and formal attire with white coat in Japan (35.3%) and the USA (45.7%). The highest rated form of attire across all respondents was formal attire with white coat (38.6%). Among all respondents, 78.6% preferred some form of attire with a white coat, while 21.4% preferred a form without a white coat. Table 3 shows preferred physician attire by physician type and clinical care setting.

## Importance, impact and appropriateness of physician attire and white coats

Respondent opinions were sought using a Likert scale in which a score of 1 indicated 'strongly disagree' and 5 'strongly agree.' In response to the prompt 'how my doctor dresses is important to me,' mean scores were similar for Italy (3.55), Japan (3.51) and the USA (3.49) and lower for Switzerland (3.05) (p<0.05 for all three pairwise comparisons). In response to the prompt 'how my doctor dresses influences how happy I am with the care I receive,' mean scores for Italy were 2.92, Japan 3.22, Switzerland 2.47 and the USA 3.17 (p<0.05 for all pairwise comparisons except for Japan-USA). In response to the prompt 'it is appropriate for a doctor to dress casually when seeing patients over the weekend,' all countries differed with mean scores for Italy of 3.15, Japan 2.57, Switzerland 3.37 and the USA 3.27 (p<0.05 for all six pairwise comparisons).

With respect to perceptions of whether white coats should be worn by physicians in various settings, differences emerged. When considering whether physicians should wear a white coat when seeing patients in their office, mean scores for Italy were 3.92, Japan 3.59, Switzerland 3.27 and the USA 3.53 (p<0.05 for all pairwise comparisons except for Japan-USA). When asked if physicians should wear a white coat in the emergency department, mean scores for Italy were 4.06, Japan 3.05, Switzerland 4.02, and the USA 3.34 (p<0.05 for all pairwise comparisons except for Italy-Switzerland). When asked if physicians should wear a white coat in the hospital, all countries differed with mean scores for Italy of 4.16, Japan 3.57, Switzerland 3.89 and the USA 3.63 (p<0.05 for all six pairwise comparisons). In response to the prompt 'doctors should always wear a white coat when seeing patients in any setting,' all countries differed with mean scores for Italy of 3.56, Japan 2.99, Switzerland 2.82

**Table 3** Preferred physician attire by physician type and care setting

| Physician type | Attire | Italy | Japan | Switzerland | USA | Total |
|---|---|---|---|---|---|---|
| Primary care physician | Casual | 103 (11.0%) | 33 (1.6%) | 199 (24.4%) | 158 (3.0%) | 493 (5.5%) |
| | Casual with white coat | 165 (17.6%) | 682 (34.1%) | 183 (22.4%) | 518 (9.9%) | 1548 (17.2%) |
| | Scrubs | 61 (6.5%) | 188 (9.4%) | 90 (11.0%) | 238 (4.6%) | 577 (6.4%) |
| | Scrubs with white coat | 126 (13.5%) | 357 (17.9%) | 78 (9.6%) | 742 (14.2%) | 1303 (14.5%) |
| | Formal | 128 (13.7%) | 49 (2.5%) | 73 (8.9%) | 787 (15.0%) | 1037 (11.6%) |
| | Formal with white coat | 296 (31.6%) | 669 (33.4%) | 188 (23.0%) | 2451 (46.8%) | 3604 (40.1%) |
| | Business suit | 57 (6.1%) | 22 (1.1%) | 6 (0.7%) | 340 (6.5%) | 425 (4.7%) |
| Emergency department physician | Casual | 36 (3.9%) | 42 (2.1%) | 31 (3.8%) | 63 (1.2%) | 172 (1.9%) |
| | Casual with white coat | 89 (9.6%) | 206 (10.3%) | 65 (8.0%) | 298 (5.7%) | 658 (7.3%) |
| | Scrubs | 343 (37.2%) | 1131 (56.5%) | 382 (46.9%) | 2108 (40.2%) | 3964 (44.2%) |
| | Scrubs with white coat | 324 (35.1%) | 354 (17.7%) | 271 (33.3%) | 1784 (34.1%) | 2733 (30.4%) |
| | Formal | 16 (1.7%) | 61 (3.0%) | 8 (1.0%) | 134 (2.6%) | 219 (2.4%) |
| | Formal with white coat | 105 (11.4%) | 204 (10.2%) | 52 (6.4%) | 793 (15.1%) | 1154 (12.9%) |
| | Business suit | 10 (1.1%) | 5 (0.2%) | 5 (0.6%) | 60 (1.1%) | 80 (0.9%) |
| Hospital physician | Casual | 25 (2.7%) | 19 (1.0%) | 33 (4.1%) | 68 (1.3%) | 145 (1.6%) |
| | Casual with white coat | 98 (10.6%) | 680 (34.0%) | 138 (17.0%) | 435 (8.3%) | 1351 (15.1%) |
| | Scrubs | 176 (19.1%) | 162 (8.1%) | 203 (25.0%) | 594 (11.4%) | 1135 (12.7%) |
| | Scrubs with white coat | 404 (43.8%) | 444 (22.2%) | 285 (35.0%) | 1600 (30.7%) | 2733 (30.5%) |
| | Formal | 17 (1.8%) | 26 (1.3%) | 20 (2.4%) | 346 (6.6%) | 409 (4.6%) |
| | Formal with white coat | 189 (20.5%) | 660 (33.0%) | 129 (15.9%) | 1964 (37.6%) | 2942 (32.8%) |
| | Business suit | 14 (1.5%) | 9 (0.4%) | 5 (0.6%) | 212 (4.1%) | 240 (2.7%) |
| Surgeon | Casual | 32 (3.5%) | 13 (0.6%) | 17 (2.1%) | 37 (0.7%) | 99 (1.1%) |
| | Casual with white coat | 85 (9.2%) | 238 (11.9%) | 44 (5.4%) | 179 (3.4%) | 546 (6.1%) |
| | Scrubs | 289 (31.2%) | 942 (47.1%) | 345 (42.6%) | 2224 (42.5%) | 3800 (42.4%) |
| | Scrubs with white coat | 302 (32.6%) | 501 (25.0%) | 272 (33.6%) | 1202 (23.0%) | 2277 (25.4%) |
| | Formal | 37 (4.0%) | 35 (1.8%) | 17 (2.1%) | 192 (3.7%) | 281 (3.1%) |
| | Formal with white coat | 155 (16.8%) | 266 (13.3%) | 108 (13.3%) | 1102 (21.1%) | 1631 (18.2%) |
| | Business suit | 25 (2.7%) | 6 (0.3%) | 7 (0.9%) | 291 (5.6%) | 329 (3.7%) |
| Overall | Casual | 20 (2.2%) | 17 (0.9%) | 46 (5.8%) | 70 (1.4%) | 153 (1.7%) |
| | Casual with white coat | 94 (10.2%) | 606 (30.3%) | 136 (17.0%) | 367 (7.1%) | 1203 (13.5%) |
| | Scrubs | 146 (15.8%) | 203 (10.1%) | 205 (25.6%) | 390 (7.5%) | 944 (10.6%) |
| | Scrubs with white coat | 385 (41.7%) | 436 (21.8%) | 252 (31.5%) | 1289 (24.8%) | 2362 (26.5%) |
| | Formal | 25 (2.7%) | 26 (1.3%) | 22 (2.7%) | 448 (8.6%) | 521 (5.9%) |
| | Formal with white coat | 235 (25.5%) | 707 (35.3%) | 131 (16.4%) | 2370 (45.7%) | 3443 (38.6%) |
| | Business suit | 18 (1.9%) | 7 (0.3%) | 8 (1.0%) | 255 (4.9%) | 288 (3.2%) |

and the USA 3.12 (p<0.05 for all six pairwise comparisons). These results are summarised in table 4 and online supplemental appendix E.

### Comparisons of patient preferences within countries

Similarities and differences were identified when comparing preferences within countries based on respondent sociodemographic characteristics. When examining respondent gender, men and women rated different types of physician attire similarly within their respective countries. The only significant difference was that men rated formal attire more highly than women in Switzerland (male composite score 6.2, female composite score 5.4, p=0.04) (online supplemental appendix F). When comparing respondents aged 65 years and older with those between 18 and 64 years, there were no significant differences in composite scores for all types of physician attire in both Italy and Switzerland. In contrast, when compared with the younger cohort, respondents aged 65 years and older rated casual attire, formal attire, formal attire with white coat and business suit more highly

**Table 4** Respondent opinions regarding importance, influence and appropriateness of physician attire and white coats

| | Italy | Japan | Switzerland | USA | Total |
|---|---|---|---|---|---|
| **How my doctor dresses is important to me.** | | | | | |
| Strongly disagree | 60 (6.4%) | 67 (3.3%) | 110 (13.4%) | 222 (4.2%) | 459 (5.1%) |
| Disagree | 87 (9.4%) | 280 (13.9%) | 151 (18.4%) | 531 (10.0%) | 1049 (11.6%) |
| Neither agree nor disagree | 220 (23.7%) | 430 (21.4%) | 260 (31.8%) | 1603 (30.2%) | 2513 (27.7%) |
| Agree | 410 (44.1%) | 1031 (51.3%) | 185 (22.6%) | 2303 (43.5%) | 3929 (43.4%) |
| Strongly agree | 153 (16.4%) | 202 (10.1%) | 113 (13.8%) | 641 (12.1%) | 1109 (12.2%) |
| Mean* | 3.55 | 3.51 | 3.05 | 3.49 | |
| **How my doctor dresses influences how happy I am with the care I receive.** | | | | | |
| Strongly disagree | 132 (14.3%) | 124 (6.2%) | 223 (27.3%) | 334 (6.3%) | 813 (9.0%) |
| Disagree | 209 (22.6%) | 396 (19.7%) | 235 (28.8%) | 851 (16.1%) | 1691 (18.7%) |
| Neither agree nor disagree | 250 (27.0%) | 536 (26.7%) | 171 (20.9%) | 2088 (39.5%) | 3045 (33.7%) |
| Agree | 263 (28.5%) | 812 (40.5%) | 124 (15.2%) | 1633 (30.9%) | 2832 (31.3%) |
| Strongly agree | 70 (7.6%) | 138 (6.9%) | 64 (7.8%) | 384 (7.2%) | 656 (7.3%) |
| Mean* | 2.92 | 3.22 | 2.47 | 3.17 | |
| **It is appropriate for a doctor to dress casually when seeing patients over the weekend.** | | | | | |
| Strongly disagree | 81 (8.7%) | 209 (10.4%) | 104 (12.8%) | 182 (3.5%) | 576 (6.4%) |
| Disagree | 213 (22.9%) | 837 (41.7%) | 139 (17.2%) | 955 (18.1%) | 2144 (23.7%) |
| Neither agree nor disagree | 218 (23.4%) | 613 (30.5%) | 147 (18.2%) | 1761 (33.3%) | 2739 (30.3%) |
| Agree | 326 (35.1%) | 300 (15.0%) | 189 (23.4%) | 2047 (38.7%) | 2862 (31.7%) |
| Strongly agree | 92 (9.9%) | 48 (2.4%) | 230 (28.4%) | 340 (6.4%) | 340 (7.9%) |
| Mean* | 3.15 | 2.57 | 3.37 | 3.27 | |
| **Doctors should wear a white coat when seeing patients in their office.** | | | | | |
| Strongly disagree | 20 (2.2%) | 48 (2.4%) | 108 (13.2%) | 84 (1.6%) | 260 (2.9%) |
| Disagree | 47 (5.1%) | 226 (11.2%) | 132 (16.1%) | 552 (10.4%) | 957 (10.6%) |
| Neither agree nor disagree | 139 (14.9%) | 437 (21.7%) | 170 (20.8%) | 1698 (32.1%) | 2444 (27.0%) |
| Agree | 504 (54.1%) | 1085 (54.0%) | 251 (30.7%) | 2361 (44.7%) | 4201 (46.4%) |
| Strongly agree | 221 (23.7%) | 214 (10.7%) | 157 (19.2%) | 593 (11.2%) | 1185 (13.1%) |
| Mean* | 3.92 | 3.59 | 3.27 | 3.53 | |
| **Doctors should wear a white coat when seeing patients in the emergency department.** | | | | | |
| Strongly disagree | 15 (1.6%) | 102 (5.1%) | 47 (5.8%) | 111 (2.1%) | 275 (3.0%) |
| Disagree | 36 (3.8%) | 541 (27.0%) | 56 (6.9%) | 828 (15.6%) | 1461 (16.2%) |
| Neither agree nor disagree | 115 (12.3%) | 623 (31.1%) | 75 (9.2%) | 1952 (36.9%) | 2765 (30.6%) |
| Agree | 480 (51.2%) | 628 (31.3%) | 294 (36.0%) | 1973 (37.3%) | 3375 (37.3%) |
| Strongly agree | 291 (31.1%) | 110 (5.5%) | 343 (42.1%) | 426 (8.1%) | 1170 (12.9%) |
| Mean* | 4.06 | 3.05 | 4.02 | 3.34 | |
| **Doctors should wear a white coat when seeing patients in the hospital.** | | | | | |
| Strongly disagree | 13 (1.4%) | 45 (2.2%) | 50 (6.1%) | 65 (1.2%) | 173 (1.9%) |
| Disagree | 19 (2.0%) | 236 (11.7%) | 45 (5.5%) | 401 (7.6%) | 701 (7.7%) |
| Neither agree nor disagree | 83 (8.8%) | 441 (22.0%) | 128 (15.7%) | 1507 (28.5%) | 2159 (23.9%) |
| Agree | 509 (54.3%) | 1114 (55.4%) | 311 (38.2%) | 2756 (52.1%) | 4690 (51.8%) |
| Strongly agree | 314 (33.5%) | 174 (8.7%) | 281 (34.5%) | 560 (10.6%) | 1329 (14.7%) |
| Mean* | 4.16 | 3.57 | 3.89 | 3.63 | |
| **Doctors should always wear a white coat when seeing patients in any setting.** | | | | | |
| Strongly disagree | 23 (2.5%) | 109 (5.4%) | 179 (21.9%) | 181 (3.4%) | 492 (5.4%) |
| Disagree | 119 (12.7%) | 567 (28.2%) | 164 (20.0%) | 1140 (21.5%) | 1990 (22.0%) |

**Table 4** Continued

| | Italy | Japan | Switzerland | USA | Total |
|---|---|---|---|---|---|
| Neither agree nor disagree | 269 (28.7%) | 682 (33.9%) | 202 (24.7%) | 2147 (40.6%) | 3300 (36.4%) |
| Agree | 361 (38.5%) | 550 (27.4%) | 169 (20.7%) | 1497 (28.3%) | 2577 (28.5%) |
| Strongly agree | 165 (17.6%) | 103 (5.1%) | 104 (12.7%) | 326 (6.2%) | 698 (7.7%) |
| Mean* | 3.56 | 2.99 | 2.82 | 3.12 | |

*Means calculated with scores of 1 assigned to 'strongly disagree,' 3 to 'neither agree nor disagree' and 5 to 'strongly agree.'

in both Japan and the USA. When compared with the younger cohort, respondents aged 65 years and older rated casual attire with white coat and scrubs more highly in Japan, a finding that was not significant in the USA (online supplemental appendix G). There was no association between respondent preferences for physician attire and number of physicians seen in the prior year.

## DISCUSSION

In this international, multicentre, cross-sectional study, we report preferences of 9171 patients for physician attire across a variety of geographic regions, clinical contexts, physician types and patient sociodemographic characteristics. We found that the highest rated form of physician attire differed across countries, but that all most strongly preferred a white coat with any attire. Respondents from the USA more strongly preferred all types of formal attire compared with those from Italy, Japan and Switzerland. All countries more strongly preferred scrubs-based attire for emergency department physicians and surgeons. Taken together, these findings suggest that how a physician dresses has importance that varies around the world.

Our study adds to the existing literature by demonstrating that patients harbour expectations of how their physicians dress and that these expectations depend on sociocultural norms, context and patient factors. In some clinical care contexts, preferences vary substantially. In others, they are nearly universal such as those for emergency department physicians and surgeons wearing scrubs-based attire. With some exceptions, patients tended to dislike extremes in attire such as casual or business suit. Finally, it was very common for patients to prefer their physicians wear a white coat, a historically traditional aspect of the physician's uniform and what is often considered a symbol of the profession.[24] This was particularly evident when patient preferences for the underlying form of attire were split (eg, primary care and hospital physicians).

Other studies exploring patient perceptions for physician attire have yielded a diverse and often conflicting array of findings, most of which are complicated by different measurement tools and outcomes. Consistent with our results, numerous studies across continents have identified a clear patient preference for white coats.[6 7 10 12 14 23 25–41] However, some studies reveal no significant preferences,[42–45] and others indicate that the

white coat may even cause higher levels of tension in patients.[44] Some studies have shown that physician attire carries little importance with patients,[46–50] whereas others have shown it has a substantial impact on the patient experience,[30 51] congruent with our results. Literature differs on whether preferences for the white coat change after patients are educated about potential risk of microbial transmission, with some studies showing decreased preference[14 52] and another showing no change.[35] Studies examining attire in countries with bare-below-the-elbow policies have indicated near universal disdain for this infection prevention measure.[27 35] Some studies have shown preference for different forms of attire such as scrubs (eg, specific circumstances such as gastroenterology suites[18 53] and emergencies[5]) and informal attire,[54] and some have revealed no specific patient preferences.[52 55 56] Five studies noted that patient perceptions of compassion, professionalism and credibility were not associated with a physician's dress.[25 32 57–59] Finally, some studies have demonstrated that attire is more important to patients who are older,[34 51 60] a finding we noted in Japan and the USA.

Studies conducted around the globe have repeatedly demonstrated that context is crucial when considering nonverbal cues like physician dress. Patient viewpoints are associated with a variety of factors such as type of care delivered, type of physician and even time of day. In one example, Switzerland has a defined healthcare uniform of white scrubs and white coat.[20] This relatively unique phenomenon likely caused patients in Switzerland to expect this form of attire and thus strongly prefer it to other forms. In another example from the USA, parents of children being evaluated in the paediatric emergency department were more likely to prefer physicians wearing scrubs but only if their children were experiencing a surgical emergency.[46] Likewise, in that same study, parents who visited the emergency department during the day shift preferred formal attire, whereas those who visited during the night preferred less formal attire.[46] Finally, preferences have also previously been shown to deviate from cultural norms or established national dress.[11 13 30 38] For instance, patients in family medicine clinics in Saudi Arabia were more likely to adhere to medical recommendations and return for subsequent care if the physician was dressed in Western garb[60]; yet this same population was significantly more willing to discuss personal issues

such as psychological problems with a physician wearing Saudi national dress.[60] This finding of preferences that varied based on topic of conversation was noted in other studies as well.[9 10]

A number of strengths distinguish our study from others that have previously investigated patient preferences for physician attire. To our knowledge, this study of over 9000 participants is the largest examination of opinions on physician dress to date. We employed a standardised survey instrument which allowed direct comparisons across diverse geography and contexts. Randomisation of photograph sequence and instrument delivery reduced the risk of ordering, priming and anchoring bias. We also used photographs containing physician models with identical postures, facial expressions, lighting and background, all of which limited the confounding associated with previous studies using models of different backgrounds and appearances.[16–18 51 61] Finally, our findings have important policy implications for physician dress code in different care settings and areas of the world.

Our study also has limitations. Our physician models were young, slender and either Caucasian or Asian, and as such were not representative of the various sociodemographic characteristics of physicians. Likewise, purposeful differences among survey instruments, including white scrubs instead of blue scrubs in the Switzerland survey and physician models of Japanese descent in the Japan survey, were introduced to ensure relevance. Our study overrepresented the USA more so than Japan and the European countries, which could have contributed to skewed results and greater power in any comparison with the USA. This was particularly evident when examining attire for hospital physicians, for example, in which the highest preference for formal attire with white coat was driven by US respondents. Despite large sample sizes in Italy and Switzerland, only one clinical site was represented in each of these countries, and this may not fully represent the country. When feasible from our convenience sampling methodology, we surveyed multiple clinical sites, because this approach likely achieved better representation of patients' preferences for different forms of attire in the respective countries. We did not obtain results from other regions including Africa, Australia, the Middle East and South America, which could have contributed noteworthy input. Countries yielded different arrays of respondent sociodemographic characteristics such as age and education, which led to disproportionate representation among some groups. The survey instrument used Likert scales with predefined categories which may not accurately reflect nuanced patient opinions, and the clinical relevance of small but significant differences in these scales is unknown. The instrument did not capture or explore other elements of etiquette-based patient–physician interaction[62] such as introductions and smiles,[17 18 26 36 45] which are known to be paramount for ensuring effective healthcare relationships. Our study did not compare the relative impacts of physician attire with these and other factors known to influence the patient–physician relationship

such as health literacy,[63] communication skills[64 65] and respect for patient autonomy.[64] Finally, the data from several of the individual country-specific studies have been previously published. However, this study is the first instance in which all data are compiled to allow for cross-national comparisons.

In conclusion, the effects of physician attire on the patient experience are complex and multilayered. Our findings suggest that one-size-fits-all physician attire policies which extend to all healthcare specialties and contexts are unlikely to reflect the desires and expectations of patients. Instead, our nuanced results that harness direct patient preferences may be used to inform local, regional and national healthcare policy-makers and leaders in their efforts to define physician uniforms. Given that preferences vary, a tailored approach should be sought that matches attire with acuity, setting and context. This approach is most likely to cultivate the patient–physician relationship and in turn enhance patient satisfaction, trust, confidence and health outcomes.

**Author affiliations**
[1]Medicine Service, VA Ann Arbor Healthcare System, Ann Arbor, Michigan, USA
[2]Department of Medicine, University of Michigan, Ann Arbor, Michigan, USA
[3]Department of Medicine, NYU Langone Health, New York, New York, USA
[4]W K Kellogg Eye Center, Ann Arbor, Michigan, USA
[5]Psychiatric University Hospital Zurich Department of Social and General Psychiatry Zurich West, Zurich, Switzerland
[6]Department of Infectious Diseases, Inselspital University Hospital Bern, Bern, Switzerland
[7]Department of Pediatrics, Niigata University Faculty of Medicine Graduate School of Medical and Dental Science, Niigata, Japan
[8]Department of General Internal Medicine, Fukushima Medical University Aizu Medical Center, Fukushima, Japan
[9]Emergency and Critical Care Center, Kurashiki Central Hospital, Okayama, Japan
[10]Department of Medicine, Muribushi Project for Okinawa Residency Programs, Okinawa, Japan
[11]Department of Experimental and Clinical Medicine, University of Florence, Florence, Italy
[12]Department of Neurosciences, Psychology, Drug Research and Child Health (NEUROFARBA), University of Florence, Florence, Italy
[13]Centre for Public Health, Queen's University Belfast, Belfast, UK
[14]Department of Medicine, University of Colorado, Denver, Colorado, USA

**Contributors** Conception and design of the work: SS, CP and VC. Acquisition of the data: SS, CP, LK, VC, MZ, HS, AK, and KK. Analysis and interpretation of the data: NH, SS, CP, LK, DR, LDL, MZ, HS, KK, AK, YT, CF, GV, SF and VC. Drafting of the manuscript: NH, CP, LK, DR, and VC. Critical revision of the manuscript for important intellectual content: NH, SS, CP, LK, DR, LDI, MZ, HS, KK, AK, YT, CF, GV, SF and VC. Accountable for all aspects of the work: NH, SS, CP, LK, DR, LDL, MZ, HS, KK, AK, YT, CF, GV, SF and VC. Approval of the final manuscript: NH, SS, CP, LK, DR, LDL, MZ, HS, KK, AK, YT, CF, GV, SF and VC. Guarantor: NH.

**Funding** This work was partially supported by a Swiss National Science Foundation grant (32 003B_149474; PI, HS). Several investigators (SS, HS, MZ, VC and LDL) received extramural funding for salary support. All authors had full access to all the data in the study and accept responsibility for the decision to submit for publication.

**Competing interests** None declared.

**Patient and public involvement** Patients and/or the public were not involved in the design, or conduct, or reporting, or dissemination plans of this research.

**Patient consent for publication** Not applicable.

**Ethics approval** The country-specific ethical review committees that reviewed and approved or deemed this project exempt from regulation were the University of Michigan Institutional Review Board (USA, HUM00085305); the Cantonal Ethics Review Board of Zurich, based on the Swiss law on research on humans (Switzerland, No. 60-2015); the ethics committee for Tokyo Joto Hospital (Japan, No. 2015-0001) and the ethics committee for Careggi University Hospital, according to the Declaration of Helsinki (Italy, CE 7123). Participants gave informed consent to participate in the study before taking part.

**Provenance and peer review** Not commissioned; externally peer reviewed.

**Data availability statement** No data are available.

**ORCID iDs**
Nathan Houchens http://orcid.org/0000-0003-3923-0711
Yasuharu Tokuda http://orcid.org/0000-0002-9325-7934
Carlo Fumagalli http://orcid.org/0000-0001-7963-5049
Gianni Virgili http://orcid.org/0000-0002-9960-2989

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
