## [Reviewer comments · BMJ Open]

ARTICLE DETAILS

TITLE (PROVISIONAL)	International Patient Preferences for Physician Attire: Results from Cross-Sectional Studies in Four Countries Across Three Continents
AUTHORS	Houchens, Nathan; Saint, Sanjay; Petrilli, Christopher; Kuhn, Latoya; Ratz, David; De Lott, Lindsey; Zollinger, Marc; Sax, Hugo; Kamata, Kazuhiro; Kuriyama, Akira; Tokuda, Yasuharu; Fumagalli, Carlo; Virgili, Gianni; Fumagalli, Stefano; Chopra, Vineet

VERSION 1 – REVIEW

REVIEWER	Hiroshi Kurihara Kasumigaura Medical Center, Department of General Medicine and Primary Care
REVIEW RETURNED	28-Feb-2022

GENERAL COMMENTS	Your research is very unique in that you have developed a tool for international comparison of physician attire. I think it is also interesting that they tried to make comparisons across multiple countries. 1. The surveys in the U.S. and Japan were conducted in multiple clinics, but are there any differences among these clinics? If there are differences, they can be explained by smaller reasons rather than national or cultural differences. For example, patients at each clinic positively perceive the doctor's attire that they are used to seeing. 2. About appendix E. Even though the sample size is large, Italy and Switzerland are hardly representative of the country with only one institution each. Aren't the conditions too different to make a comparison between nations? Rather than differences between countries, it could simply be considered as differences between each clinic. A comparison between two countries of each participating country may not be very meaningful, but what do you think about it? 3. In your study, Italy and Switzerland have a lower percentage of people over 65 than the US and Japan. In those countries, are there any differences in the age groups other than "65 and over"? 4. Referring to Appendix C, it appears that wearing a white coat, whether casual, scrubs, or formal, all styles, was rated higher by the subjects. While it is important to know what type of clothing is preferred, I think it would be an interesting finding if wearing a white coat increases ratings.
---

	I consider this information to be useful in showing impressions of a white coat. If there is a statistically significant difference, even better, why not add it to the results and discussion of your paper?
--	--

REVIEWER	Franz Porzsolt University of Ulm, General and Vic'sceral Surgery
REVIEW RETURNED	04-Mar-2022

GENERAL COMMENTS	The project addresses an interesting but also rather complex question. I'm not sure that the "physician attire is an important element in establishing patient confidence and trust ...". There are many other attitudes that may be more important to patients than the physician attire. Unless the more important variables can be quantified it will be difficult to interpret less important factors. Second, the experimental study design requires the exact definition of the study conditions e.g., of the interventions that are expected to influence the assessed endpoints and the definition of confounders as exclusion criteria. The general concept of the project is interesting but may benefit from a critical analysis of the strength of factors that influence the doctor-patient relationship.
---

VERSION 1 – AUTHOR RESPONSE

Reviewer: 1

Your research is very unique in that you have developed a tool for international comparison of physician attire. I think it is also interesting that they tried to make comparisons across multiple countries.

1. The surveys in the U.S. and Japan were conducted in multiple clinics, but are there any differences among these clinics? If there are differences, they can be explained by smaller reasons rather than national or cultural differences. For example, patients at each clinic positively perceive the doctor's attire that they are used to seeing.

Response: We thank the reviewer for their comment. We agree that patients' preferences for physician attire are based in what they are used to seeing, not just in actual clinical sites of care but also as represented in popular media such as social media, television, movies, books, and magazines. We did not collect granular data from individual sites (i.e., comparing two clinic sites within the same geographic region of a country). We chose to focus on general comparisons between countries to highlight similarities and contrasts in patient preferences across geographic locales. Additionally, we only examined patient preferences for physician attire based on photographs of dress. We did not assess the actual attire physicians wore at each clinic site in any country nor the individual sites' dress codes/policies: therefore, we believe that the preferences patients indicated are reflective of actual attire/expectations in that clinical location plus those influenced by the culture/expectations of that area. Our research question was therefore focused on understanding variations across regions, inclusive of variation that may be brought by differences in clinics across a country.

2. About appendix E. Even though the sample size is large, Italy and Switzerland are hardly representative of the country with only one institution each. Aren't the conditions too different to make a comparison between nations? Rather than differences between countries, it could simply be considered as differences between each clinic. A comparison between two countries of each participating country may not be very meaningful, but what do you think about it?

Response: As noted in our response to item 1, we are not able to provide more granular data between individual clinic sites without leading to remarkable complexity that would hinder interpretation by *BMJ Open* readers. In the design of these studies, the research team purposefully did not combine Italy and Switzerland into one group, since these two countries are markedly different with respect to population and culture. We pursued convenience sampling in both of these countries, which did not allow for inclusion of multiple clinics. When drawing comparisons among countries, even though there are regional intra-country differences in preferences, we chose to examine preferences across countries. If the reviewer is suggesting that we remove data from all but one clinic site in the US and Japan (to compare a single clinical site in each country), this would limit our examination of as many regions as possible. This change would also decrease our total *n* by orders of magnitude and thus diminish the study's power being the largest review of patient preferences for physician attire. To address this important point, we have added the following statements to our Discussion section limitations paragraph:

“Despite large sample sizes in Italy and Switzerland, only one clinical site was represented in each of these countries, and this may not fully represent the country. When feasible from our convenience sampling methodology, we surveyed multiple clinical sites because this approach likely achieved better representation of patients' preferences for different forms of attire in the respective countries.”

3. In your study, Italy and Switzerland have a lower percentage of people over 65 than the US and Japan. In those countries, are there any differences in the age groups other than "65 and over"?

Response: We thank the reviewer for their comment. Indeed, as mentioned in the manuscript text, the demographic profiles of Italy and Switzerland are different than Japan and the US. To fully portray preferences for all age groups, we have expanded Appendix G with each age range depicted. In response to the specific reviewer comment, there were no differences noted across age groups in either Italy or Switzerland.

4. Referring to Appendix C, it appears that wearing a white coat, whether casual, scrubs, or formal, all styles, was rated higher by the subjects. While it is important to know what type of clothing is preferred, I think it would be an interesting finding if wearing a white coat increases ratings. I consider this information to be useful in showing impressions of a white coat. If there is a statistically significant difference, even better, why not add it to the results and discussion of your paper?

Response: We thank the reviewer for their suggestion and agree with drawing more attention to comparisons between forms of attire with white coat and those without white coat. We have revised the manuscript in the following ways to more effectively draw the reader to these comparisons.

1. Within the Results section "Ratings of Attire Types by Country," we have included the following new statement:
"Within each country, composite scores for attire forms with white coat were higher than those for the corresponding forms without white coat, with only one exception (composite scores for scrubs and scrubs with white coat in Japan were 6.8 and 6.6, respectively)."
2. Within the Results section "Comparisons of Patient Preferences Between Countries / Preferences for Physician Attire by Type of Physician," we have included the following new statement:
"Among all respondents, 78.6% preferred some form of attire with a white coat, while 21.4% preferred a form without a white coat."

For #2, the p value is < 0.001 . While clearly statistically significant, our statistician suggests that we do not include the p value, because it does little to add to the existing narrative (with key data already portrayed). Within the first paragraph of the Discussion section, we believe the following succinct statement highlights these important findings, and thus, it is unchanged: "We found that the highest rated form of physician attire differed across countries, but that all most strongly preferred a white coat with any attire."

Reviewer: 2

1. The project addresses an interesting but also rather complex question. I'm not sure that the "physician attire is an important element in establishing patient confidence and trust ...". There are many other attitudes that may be more important to patients than the physician attire. Unless the more important variables can be quantified it will be difficult to interpret less important factors.

Response: We agree with the reviewer that physician attire is only one of the many myriad factors that affect the doctor-patient relationship. A review of the literature reveals multiple other variables that exert influence (including but not limited to the various aspects of verbal and nonverbal communication between two parties; perceived empathy; physician technical ability; patients' perceived level of involvement and locus of control in healthcare; perceived level of doctor facilitation in patient healthcare; mental health factors including depression levels; health literacy level; and media representation of healthcare institutions and physicians), although most studies do not compare all known variables, making direct comparison and assessment of strength of individual factors challenging. Indeed, the complexity of all elements that form the patient-physician relationship cannot be overstated. Despite this limitation, our research team and many others have shown that attire does matter to patients, and therefore we believe this topic is of import. At the end of this letter, we have included a non-exhaustive list of peer-reviewed publications that demonstrate the importance/impact of physician attire (including trust, confidence, and willingness of patients to return for follow-up care and disclose important health-related information to their physicians based on dress) along with one key point from each study.

2. Second, the experimental study design requires the exact definition of the study conditions e.g., of the interventions that are expected to influence the assessed endpoints and the definition of confounders as exclusion criteria.

Response: This study involved administering standardized photographs of models wearing various forms of attire to patients in various clinic sites. The patients were asked to rate preferences for the physician based on the attire they observed. There is no intervention or endpoint being examined other than patient preference of the attire. We are simply demonstrating statistical significance of various choices made by patients in the analysis of this paper.

3. The general concept of the project is interesting but may benefit from a critical analysis of the strength of factors that influence the doctor-patient relationship.

Response: As discussed in our response to item 1 above, we agree with the reviewer that the number of items that inform and influence the doctor-patient relationship (and subsequently patient satisfaction) are numerous, with attire being only one. However, we and other study teams have identified attire as an easily modifiable factor that impacts patient perceptions, their experiences, and the doctor-patient relationship. While every patient encounter is unique, our goal with this study was to better understand overall patient perceptions of physician attire using a more standardized approach in a geographically diverse and larger sample size than had previously been studied.

VERSION 2 – REVIEW

REVIEWER	Hiroshi Kurihara Kasumigaura Medical Center, Department of General Medicine and Primary Care
REVIEW RETURNED	03-Jul-2022
GENERAL COMMENTS	The issues that I pointed out as needing to be corrected in my last peer review have been corrected. I consider it acceptable for adoption.